# A good tennis player does not lose matches. The effects of valence congruency in processing stance-argument pairs

Naomi Kamoen*, Maria Baukje Johanna Mos

Department of Communication and Cognition, Tilburg University, Tilburg, the Netherlands

* n.kamoen@uvt.nl

**Data Availability Statement:** The data is available via Dataverse; the handle is https://hdl.handle.net/10411/SFFENZ.

**Funding:** The author(s) received no specific funding for this work.

## Abstract

According to the principle of Argumentative Orientation (AO), speakers and writers adjust their frame choice to the opinion they want to convey and hearers and readers are sensitive to this profile choice. In three reaction time studies ($N = 68$; $N = 97$; $N = 60$) we investigated whether, in line with AO, stance-argument pairs congruent in valence are easier to process and to verify than incongruent pairs. Second, we tested whether, in line with predictions from the Markedness Principle (MP), positive congruent pairs are easier to process than negative congruent pairs. In line with AO, participants made faster and more accurate judgments of congruent pairs than of incongruent pairs. This effect was observed when controlling for word length and word frequency, and occurred irrespective of the distance between the evaluative word in the stance and argument. No unambiguous effect of Markedness was found.

## Introduction

Say that you want to describe your favourite tennis player's performance. Would you say that she had lost 2 out the last 10 games, or won 8 of them? Seen from a purely descriptive perspective, this choice is arbitrary because these options describe the same reality: the truth of the one statement entails the truth of the other statement. Consequently, a winning rate of 80% (positive frame) implies a losing rate of 20% (negative frame), and vice versa.

Although these positive and negative descriptions may be truth-conditionally equivalent, research has shown time and again that the frame choice affects people's evaluations of the object described: as compared to negative wordings, positive wordings lead to a more positive evaluation of the attitude object (for an overview, see [1]). This effect generalizes across a variety of linguistic frames and a range of attitude objects: not only are tennis players evaluated more positively if their winning rate rather than their losing rate is mentioned [2], but it has also been shown that ground beef tastes better when it is described as 75% lean rather than 25% fat [3], that the performance of basketball players is evaluated better when the percentage of shots hit rather than missed is listed [4], and that a project team receives higher evaluations after reading about their success rather than their failure rate [5]. This effect is called the valence consistent shift (e.g., [1], [2], [6], [7]).

**Competing interests:** The authors have declared that no competing interests exist.

Previous research has suggested several cognitive explanations for the valence consistent shift (e.g., [1], [2], [6], [7]); see section 1.1 for an elaboration), but empirical evidence for these explanations is both scarce and primarily obtained from discourse completion tasks (an exception is the work by [8], whose study focusses on descriptive rather than evaluative sentences). These studies typically require participants to reflect consciously on the words they would use in a certain context (e.g., given a positive or negative valenced stance, what argument would you choose: a positive or a negative one?). As this is a slightly artificial way to measure cognitive mechanisms, we set out to test explanations for the valence consistent shift using reaction times. The advantage of using such a method is that it allows for measuring potential facilitating effects of certain word choices of which language users may be unaware. The paragraphs below discuss the various cognitive explanations for the valence consistent shift in more detail and describe how reaction times can be used to test their predictions.

## Cognitive mechanisms underlying the valence consistent shift

Sher and McKenzie (e.g., [6], [7], also see [8]) were among the first who formulated a cognitive explanation for the valence consistent shift. They argue that this shift is caused by differences in the reference point the speaker highlights when uttering a positive versus a negative object description, which, in turn, results in a different interpretation of the object described on the part of the hearer. We unpack Sher and McKenzie's reasoning below.

A reference point is "a cognitively salient item that gives mental access to a less salient target" ([9], pp. 34). For example, when uttering the sentence "the girl's neck" the speaker uses "the girl" as a reference point to single out a specific part of her (in this case: the neck). In natural language use, speakers continuously employ the context and the salient objects in it to tailor their choice of words such that hearers understand what they mean. This use of the term reference point is closely related to what Langacker [10] describes as the *base* against which something is *profiled*. Perhaps confusingly, he also uses the term *reference point*, but with a slightly different meaning. We refer to [2] for a discussion.

Sher and McKenzie [6] argue that the choice for positive or negative wording highlights information in a given discourse context in a similar way (compare Langacker's Current Discourse Space, or CDS, [11], pp. 59). For example, when a speaker selects a positive frame (the glass is half-full) over a negative frame (the glass is half-empty), this is because the property of the selected frame has increased relative to some previous state. Imagine a glass of wine that has just been filled after it was empty (reference point). In that case, the glass will probably be described to be "half full", because the proportion of "fullness" has recently increased relative to the baseline. Similarly, a glass of wine that was full before (reference point), but that has just been sipped halfway, will probably be described to be "half empty" as the proportion of "emptiness" has increased. In several experimental studies ([6] and [8]), Sher and McKenzie show that speakers do indeed choose a frame that reflects the property that has increased relative to a previous situation.

To explain, then, why a valence consistent shift arises, [6] argue that the hearers pick up the change with respect to the reference point that is foregrounded in the frame selected by the speaker. Therefore, hearers know that "a half full glass" is a glass that possesses quite a lot of "fullness" in the eyes of the speaker. In [6] it has been explained that this, in turn, leads to a valence consistent shift (pp. 482): "...since it is generally good to have more of a good thing, and bad to have more of a bad thing, the reference point hypothesis predicts that proportions couched in terms of good things will lead to more favorable evaluations than proportions couched in terms of bad things".

Sher and McKenzie's explanation for the valence consistent shift has been criticized in [2] because it "is not really communicative, as it does not include any reference to the communicators' intentions" and for suggesting that it is "as if associations are activated rather mechanically" (p. 2206). According to the authors, there are various argumentative discourse contexts in which speakers want to convince their audience of a certain stance; in these situations, frame-choices are chosen non-mechanically and perhaps even deliberately.

To reflect those ideas, [2] have proposed a speaker's maxim and recipient's corollary called the principle of Argumentative Orientation (AO). This principle predicts that in argumentative discourse contexts speakers tailor their frame choices such that they are in line with the evaluative direction of the stance they aim to defend. Hence, going back to the tennis player example, AO predicts if speakers foreground the "winning rate" rather than the "losing rate" of the tennis player, they are likely to do so because they want to make an argument that the player possesses some positive quality (she is a good tennis player). AO also says that hearers pick up on these hidden cues about the speaker's stance, and adjust their judgment accordingly. This, in turn, gives a more specific explanation for the occurrence of a valence consistent shift in certain argumentative contexts.

In a series of discourse completion experiments [2], AO was tested by asking participants to continue sentences like "For tennis pro Melle van Gemerden, 2005 was a *good* (positive condition) / *bad* (negative condition) year. He (a) *won* 2 / (b) *lost* 5 of his 7 international tournament matches", by picking either option (a) or (b). The authors expected—in line with AO—that positively phrased stances would lead to the selection of positively framed arguments, whereas the negatively phrased stance would more often lead to a negatively framed argument. Their studies indeed show strong evidence for AO, but a question arising from their studies is whether AO also affects unconscious processing of stance-argument pairs; in natural language use, speakers usually do not consciously reflect on their wording choices but select words in a split-second. The authors already note themselves that a next step could therefore be to investigate AO using reaction times (p. 2217). In the current study, we therefore use reaction times to investigate the prediction from AO that a positive stance (*good*) can best be defended using a positively framed argument (*win*), whereas a negative stance (*bad*) is best defended using a negative argument (*lose*). In other words, our first hypothesis states that stance-argument pairs that are congruent in their valence (*good-win* or *bad-lose*) will be verified faster and more accurately than incongruent pairs (*good-lose* or *bad-win*).

A second addition of [2] to the work of Sher and McKenzie (as reported in [6], [7], and [8]) is the introduction of the Markedness Principle (MP) within the domain of valence framing. It is well known in linguistics (e.g., see [12], [13]) that in pairs of opposites (*full/empty*, *pass/fail*, *miss/hit*), one of the members is often the neutral, unmarked term, whereas the other one is the less neutral, marked term. The positive member of a pair of opposites is usually the unmarked member. This term can be used for asking neutral questions (e.g., *how full is your glass*?), it gives name to the pair as a whole (e.g., '*fullness*'), and it (therefore) usually has a higher word frequency in language use (e.g., [12], [13], [14]). The marked term, by contrast, is only used when situational circumstances provide a reason to deviate from the default. For example, the question "How empty is your glass?" would probably be asked when someone has the intention to refill it. If the positive term is the relatively unmarked part of a contrast pair, and markedness implies an increase in cognitive load, there should be a difference in processing speed between congruent positive pairs (*win-good*) and negative pairs (*lose-bad*), with the latter being more 'costly' to process. The second hypothesis to be tested in the current research, then, is that congruent negative pairs are processed more slowly and less accurately than congruent positive pairs.

## Method Study 1

### Design and materials

To test the hypotheses, we conducted three reaction time studies in ePrime (see [15]). At the time our studies were conducted (Fall 2015, Fall 2016, Winter 2018), the General Data Protection Regulation (GDPR) had not yet entered into force. Because our studies did not include participants that are part of a vulnerable population, approval by an ethical committee was not considered necessary at the time before fielding the studies, but all studies were approved by the coordinator of the participant pool for our program. Participants did sign a consent form in which they indicated that we could use their responses for scientific research. The consent forms as well as the anonymized dataset can be accessed via Dataverse (https://hdl.handle.net/10411/SFFENZ).

In experiment 1, participants saw stance-argument pairs on a computer screen. Each pair consisted of two sentences, the first one describing a stance ("*He is a good tennis player*") and the second one describing an argument ("*He has won 20 out of the last 25 matches*"). The task for the participants was to judge if the stance and argument followed each other logically. Study 1 has also been reported on in Dutch [16].

After a practice session in which participants judged 8 stance-argument pairs, participants judged 120 experimental stance-argument pairs. For half of these items, the stance followed logically from the argument ('true pairs'"; e.g., "*He is a good tennis player. He has won 20 out of the last 25 matches*"; see the top half of Table 1), and for the other half of the items this was not the case ('false pairs'; e.g., "*This tennis player is good. He has lost 20 out of the last 25 matches*"; see the bottom half of Table 1). These false pairs were included to make sure there was variation in whether the stance followed from the argument or not. Without this variation, participants could end up responding without processing the content of the stimuli (also see guidelines for reaction time experiments by [17]). On average, participants needed 12 minutes and 31 seconds to complete the practice session and main experiment (range between 10:22 and 15:05 minutes).

**Table 1. Example of experimental materials.**

|  | Logical match | Stance Valence | Argument Valence |
|---|---|---|---|
| *Items used for testing the hypotheses* |  |  |  |
| He is a good tennis player. He has won 20 out of the last 25 matches | True | Positive | Positive |
| He is a good tennis player. He has lost 5 out of the last 25 matches | True | Positive | Negative |
| He is a bad tennis player. He has lost 20 out of the last 25 matches | True | Negative | Negative |
| He is a bad tennis player. He has won 5 out of the last 25 matches | True | Negative | Positive |
|  |  |  |  |
| *Items included to prevent automatic responding* |  |  |  |
| He is a good tennis player. He has lost 20 out of the last 25 matches | False | Positive | Negative |
| He is a good tennis player. He has won 5 out of the last 25 matches | False | Positive | Positive |
| He is a bad tennis player. He has won 20 out of the last 25 matches | False | Negative | Positive |
| He is a bad tennis player. He has lost 5 out of the last 25 matches | False | Negative | Negative |

As the hypotheses relate to true, rather than false, stance-argument pairs, only these pairs were used to test the hypothesis (see S1 Appendix for the full list of materials). The 60 true pairs related to 15 scenarios, that each occurred in 2 (stance valence: Positive or Negative) x 2 (Argument valence: Positive or Negative) variations. This way, there were 30 congruent stance-argument pairs (for which the valence of both stance and argument were the same) and an equal number of incongruent pairs (for which the valence of stance and argument were different). All participants judged all of the 120 stance-argument pairs, so a within-person design was implemented. To prevent order-effects, items were presented in different randomized orders for each participant.

## Participants

Participants (*N* = 68) were students in Communication and Information Sciences (CIS) at Tilburg University. They received partial course credit for participation. Most of the participants were female (*N* = 55) and the mean age was 21.2 (*SD* = 2.7). This sample is representative for the student population at Dutch Humanities faculties.

## Procedure

Participants were welcomed at the Communication and Information Sciences Research Laboratory at Tilburg University, where they were seated in front of a laptop. The experimenter then gave a short introduction, explaining that participants had to judge whether the stance followed logically from the argument (true or false) as quickly and accurately as possible. Next, participants saw four example items for which it was indicated whether the stance followed logically from the argument (true or false). Thereafter participants practiced with another eight items for which they had to indicate the truth-value themselves. Participants could express their judgment by pressing on a green (true) or red (false) button on the keyboard. The experimenter allowed participants to ask questions and checked if they judged the practice items correctly, which was always the case. Next, the experiment started in which participants judged all 120 stance-arguments pairs. After completing the judgment task, participants were asked to fill out a short survey with questions about their demographics.

## Analyses

In this study, there are two dependent variables. First, Accuracy is a binary dependent variable indicating if the respondent correctly judged the sentence-argument pair to be either "true" or "false". This score is predicted in a Logit model, in which one mean accuracy score is estimated for each of the four experimental conditions. These scores are allowed to vary between persons (one person may be more accurate than another person), items (one item may be easier to judge than another item) and the combination between those two (one person may make more mistakes for one item and another person for another item). The person-and item-variance are estimated at the same time, so a cross-classified multi-level model is in operation. Such a model has several advantages as compared to traditional ANOVAs (see [18] and [19]).

The second dependent variable, Reaction Time (hereafter RT), indicates the time a participant needed to respond to each stance-argument pair. To analyze such an interval variable, one traditionally uses models assuming that the scores on the dependent variable follow a normal distribution. In our data, however, the reaction times did not follow a normal distribution (this occurs frequently in eye-tracking and reaction time studies, see [20] and [21]). To correct for this, a loglinear transformation was applied. This correction method is common in eye-tracking and reaction times studies (compare [22]). Next, a multi-level model was constructed similar to the model used for analyzing the accuracy scores. The model estimates four mean

RTs (for each of the four conditions), that are allowed to vary between persons, items, and the combination of the two. Again the person and item variance are estimated simultaneously, which means that a cross-classified model is in operation. See S2 Appendix for a formalization.

For both the reaction times and the accuracy scores, our hypothesis can be tested by comparing the four cell means in a contrast test. If an interaction is found between the valence of the stance and the valence of the argument, this interaction can be specified further by checking if for negative stances there is a difference between positive versus negative arguments, as well as for the positive stances. An AO effect would occur if both congruent pairs are processed faster and responded to more accurately than their incongruent counterparts, resulting in a classic interaction effect. An effect of MP can be shown if the positive congruent pairs are processed faster and more accurately than the negative congruent pairs. MP does not make any predictions about (differences between) incongruent pairs.

## Results Study 1

### Accuracy

Fig 1 displays the mean accuracy scores for the four conditions.

Results show a main effect for stance valence ($\chi^2$ = 12.07; df = 1; $p < .001$; Cohen's $d = 0.29$): positive stances are more often judged accurately than negative stances. A similar main effect was observed for argument valence ($\chi^2$ 13.83; df = 1; $p < .001$; Cohen's $d = 0.28$): positive arguments are judged more accurately than their negative counterparts. These main effects, however, should be interpreted in light of an interaction effect between stance and argument ($\chi^2$ = 87.53; df = 1; $p < .001$). The interaction reads that positive stances are more

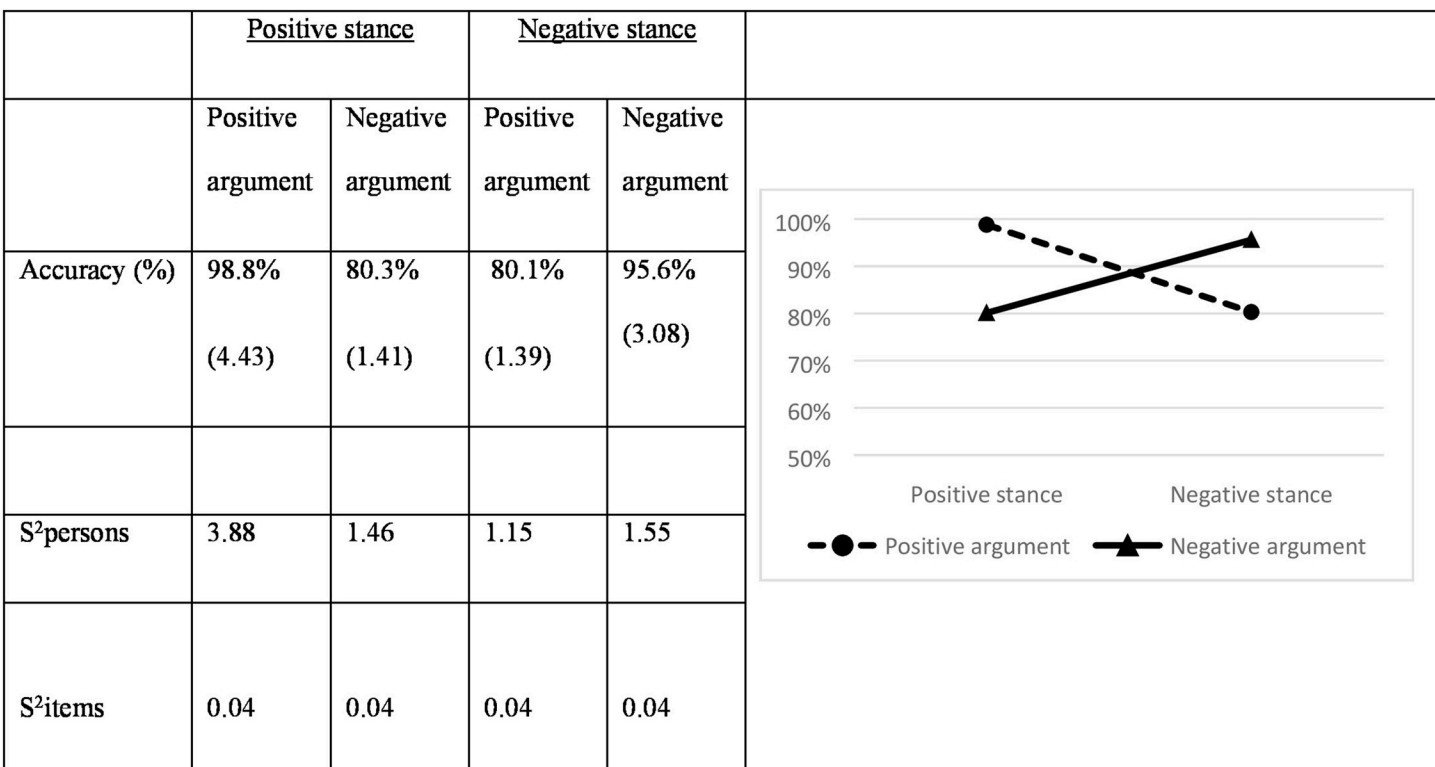

| | Positive stance | | Negative stance | | |
|---|---|---|---|---|---|
| | Positive argument | Negative argument | Positive argument | Negative argument | |
| Accuracy (%) | 98.8% (4.43) | 80.3% (1.41) | 80.1% (1.39) | 95.6% (3.08) | |
| $S^2$persons | 3.88 | 1.46 | 1.15 | 1.55 | |
| $S^2$items | 0.04 | 0.04 | 0.04 | 0.04 | |

**Fig 1. Mean accuracy scores for the four conditions (Logits between brackets) and their variances (in Logits).**

accurately judged when they are followed by a positive argument ($\chi^2$ = 67.80; df = 1; $p <$ .001; Cohen's $d$ = 1.14), whereas negative stances are judged more accurately when they are followed by a negative argument ($\chi^2$ = 49.87; df = 1; $p <$ .001; Cohen's $d$ = 0.88). In other words, congruence in valence is the key determiner in the accuracy of judgments.

To test for effects of MP, we compared the accuracy scores for the congruent positive and the congruent negative pairs. Results show a large effect ($\chi^2$ = 67.81; df = 1; $p <$ .001; Cohen's $d$ = 0.51), indicating that argument-stance pairs consisting of two positive parts are more accurately judged than pairs consisting of the negative parts.

### Reaction times

The mean RTs and corresponding variances are provided in Fig 2.

The results for the reaction times show the same pattern as the accuracy scores. Judging a positive stance and a positive argument goes faster than judging a negative stance and a negative argument (for stances: $\chi^2$ = 48.92; df = 1; $p <$ .001; Cohen's $d$ = 0.19 and for arguments $\chi^2$ = 26.92; df = 1; $p <$ .001; Cohen's $d$ = 0.14). Again, there is a much stronger interaction effect between stance and argument valence ($\chi^2$ = 198.67; df = 1; $p <$ .001), such that positive stances are judged faster when they are followed by a positive argument ($\chi^2$ = 180.81; df = 1; $p <$ .001; Cohen's $d$ = 0.69), whereas negative stances are judged faster when they are followed by a negative argument ($\chi^2$ = 62.60; df = 1; $p <$ .001; Cohen's $d$ = 0.35).

To assess the effect of MP, we compared the positive and negative congruent pairs. Similar to the accuracy scores, an effect in the expected direction was observed: positive congruent pairs are judged faster than negative congruent pairs ($\chi^2$ = 80.01; df = 1; $p <$ .001; Cohen's $d$ = 0.39).

### Conclusion and Discussion Study 1

Both the accuracy scores and the RT show a pattern that is in line with predictions from the principle of Argumentative Orientation [2]: congruent stance-argument pairs are verified faster and more accurately than incongruent pairs. Along with this interaction we observed

| | Positive stance | | Negative stance | | |
|---|---|---|---|---|---|
| | Positive argument | Negative argument | Positive argument | Negative argument | |
| Reaction time | 2990 (8.00) | 3964 (8.29) | 4113 (8.32) | 3537 (8.17) |  |
| | | | | | |
| $S^2$persons | 0.06 | 0.06 | 0.05 | 0.07 | |
| $S^2$items | 0.01 | 0.01 | 0.01 | 0.01 | |
| $S^2$pers.*item | 0.17 | 0.18 | 0.18 | 0.17 | |

**Fig 2. Mean reaction times in milliseconds (Logs between brackets) and their variances (in Logs).**

that congruent positive pairs (*good*, *win*) are verified even more accurately and faster than congruent negative pairs (*bad*, *lose*). This finding is in line with MP predicting that processing marked terms (such as *bad* and *lose*) is more cognitively demanding than processing unmarked terms.

While the faster and more accurate processing of positive congruent pairs compared to negative congruent pairs may indeed be an effect of markedness, there is a potential confound in Study 1: word frequency. The stimuli were controlled for word length (arguments: $t(14) = 0.64$; $p = .54$ and standpoints: $t(14) = 0.37$; $p = .72$), but not for word frequency: the positive standpoints and arguments were more frequent than their negative counterparts ($t(14) = 3.36$; $p = .01$ and $t(14) = 4.42$; $p = .001$ respectively). As word frequency is a factor that is known to affect reaction times (e.g., [14], [23]), this difference between conditions may explain the observed pattern, rather than be an effect of the Markedness Principle. Therefore, a second study was conducted in which the valence terms were controlled for both word length and frequency.

## Method Study 2

### Design

Study 2 used the same experimental design and the same procedure as Study 1, but different stimulus materials: again, participants were shown a stance-argument pair and were asked to judge as quickly and accurately as possible whether the argument followed logically from the stance provided. This method section therefore only elaborates on what is different from Study 1: the participants and the experimental items used in Study 2.

### Materials

Materials were similar to those used in Study 1, but this time participants judged only 6 scenarios in 2 (stance valence: positive or negative) x 2 (argument valence: positive or negative) conditions, e.g. "the festival is fun/dull. 80/20% of the visitors are amused/bored". Items were shown in individually randomized orders. The positive and negative evaluative terms were comparable with respect to both word length and word frequency based on the Log frequencies corpus SoNaR [24], a 500 million Dutch corpus (in all cases: $t(5) < 1.00$; $p > .36$; see S3 Appendix for the translated stimulus materials). Study 2 was also reported on in Dutch in [16]. There are minor differences in the analyses reported on here compared to the ones in [16], see S3 Appendix. These have not caused any changes in the observed pattern of results.

### Participants

Similar to Study 1, the participants ($N = 97$) were students at Tilburg University, who received partial course credit for participation. Their mean age was 21.2 years ($SD = 2.5$), and most participants were female ($N = 74$). None of the participants of Study 2 had participated in Study 1.

## Results Study 2

### Accuracy

Fig 3 displays the accuracy scores for each of the four experimental conditions.

There is a main effect for argument valence ($\chi^2 = 8.06$; df = 1; $p < .01$; Cohen's $d = 0.33$): in line with Study 1, pairs with positive arguments are more often correctly judged than pairs with a negative argument. Analyses also show a main effect of stance valence again ($\chi^2 = 4.15$; df = 1; $p = .04$; Cohen's $d = 0.24$), but in contrast to the results for Study 1, negative stances are judged more accurately than positive stances. These main

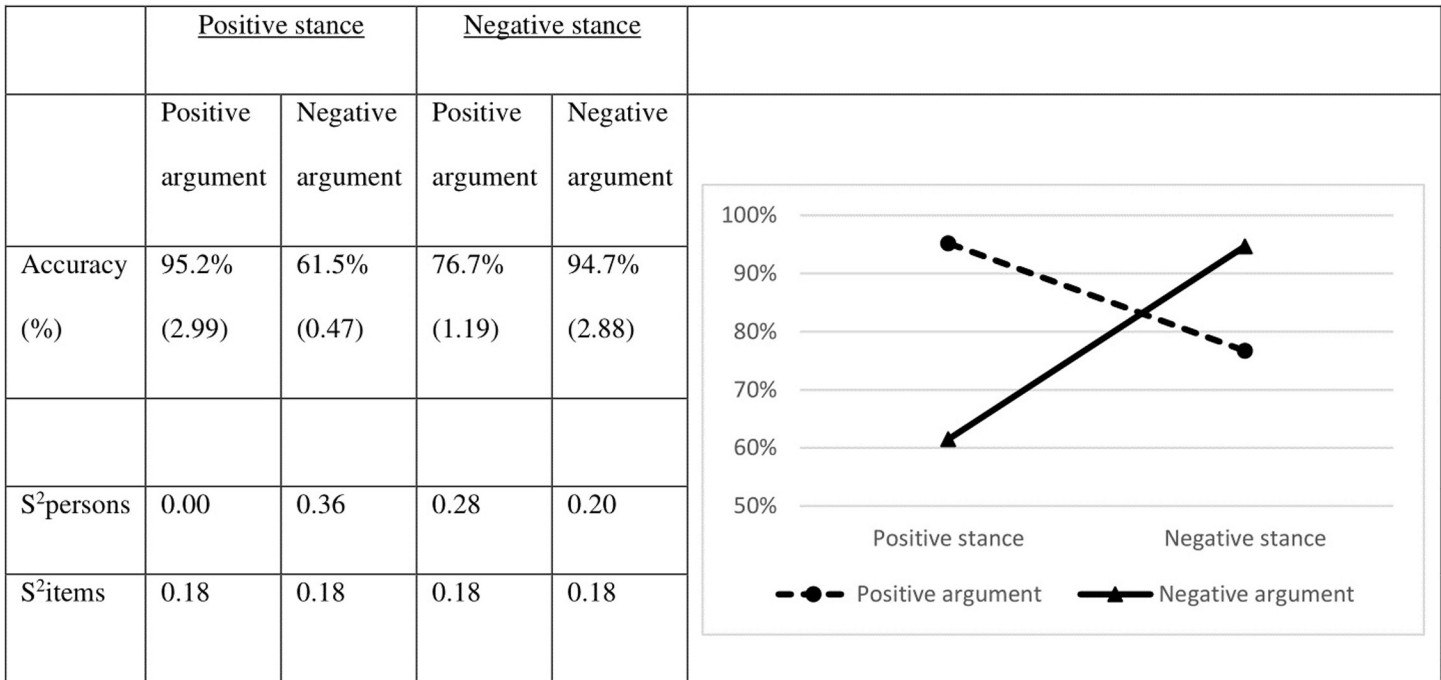

|  | Positive stance | | Negative stance | |
|---|---|---|---|---|
|  | Positive argument | Negative argument | Positive argument | Negative argument |
| Accuracy (%) | 95.2% (2.99) | 61.5% (0.47) | 76.7% (1.19) | 94.7% (2.88) |
| $S^2$persons | 0.00 | 0.36 | 0.28 | 0.20 |
| $S^2$items | 0.18 | 0.18 | 0.18 | 0.18 |

**Fig 3. Mean accuracy scores per condition in percentages (Logits between brackets) and their variances (in Logits).**

effects, however, should again be seen in light of an interaction between stance and argument valence ($\chi^2$ = 185.12; df = 1; $p$ < .001). The interaction is such that positive stances are more often correctly judged when they are backed-up by a positive argument ($\chi^2$ = 130.63; df = 1; $p$ < .001; Cohen's $d$ = 2.64), whereas negative stances are more often correctly judged when accompanied by a negative stance ($\chi^2$ = 68.55; df = 1; $p$ < .001; Cohen's $d$ = 1.59). In line with Study 1, congruent stance-argument pairs are more often judged correctly than incongruent pairs.

To test the effect of MP, we next compared the accuracy scores for the positive and negative congruent pairs. In contrast to Study 1, we did not observe an effect of MP ($\chi^2$ = 0.16; df = 1; $p$ = .69). In other words, the congruent positive pairs are judged as accurately as congruent negative pairs.

### Reaction times

The mean RT for each condition is provided in Fig 4. The pattern observed for the reaction times is similar to Study 1: judging positive stances and arguments goes faster than judging negative stances (in both cases $\chi^2$ > 4.88; df = 1; $p$ < .02; Cohen's $d$ < 0.08). Like in the previous analyses, there is an interaction between stance and argument valence ($\chi^2$ = 124.51; df = 1; $p$ < .001). Again, the congruence between stance and argument valence strongly affects RT: positive stances are judged faster when accompanied by a positive argument ($\chi^2$ = 109.77; df = 1; $p$ < .001; Cohen's $d$ = 0.49), whereas negative stances are judged faster when they are backed up by a negative argument ($\chi^2$ = 38.71; df = 1; $p$ < .001; Cohen's $d$ = 0.29).

Similar to our results in Study 1, we observed that verifying a congruent positive statement goes faster than verifying a congruent negative statement ($\chi^2$ = 16.39; df = 1; $p$ < .001; Cohen's $d$ = 0.20). Hence, for the reaction times we did find an MP effect similar to Study 1.

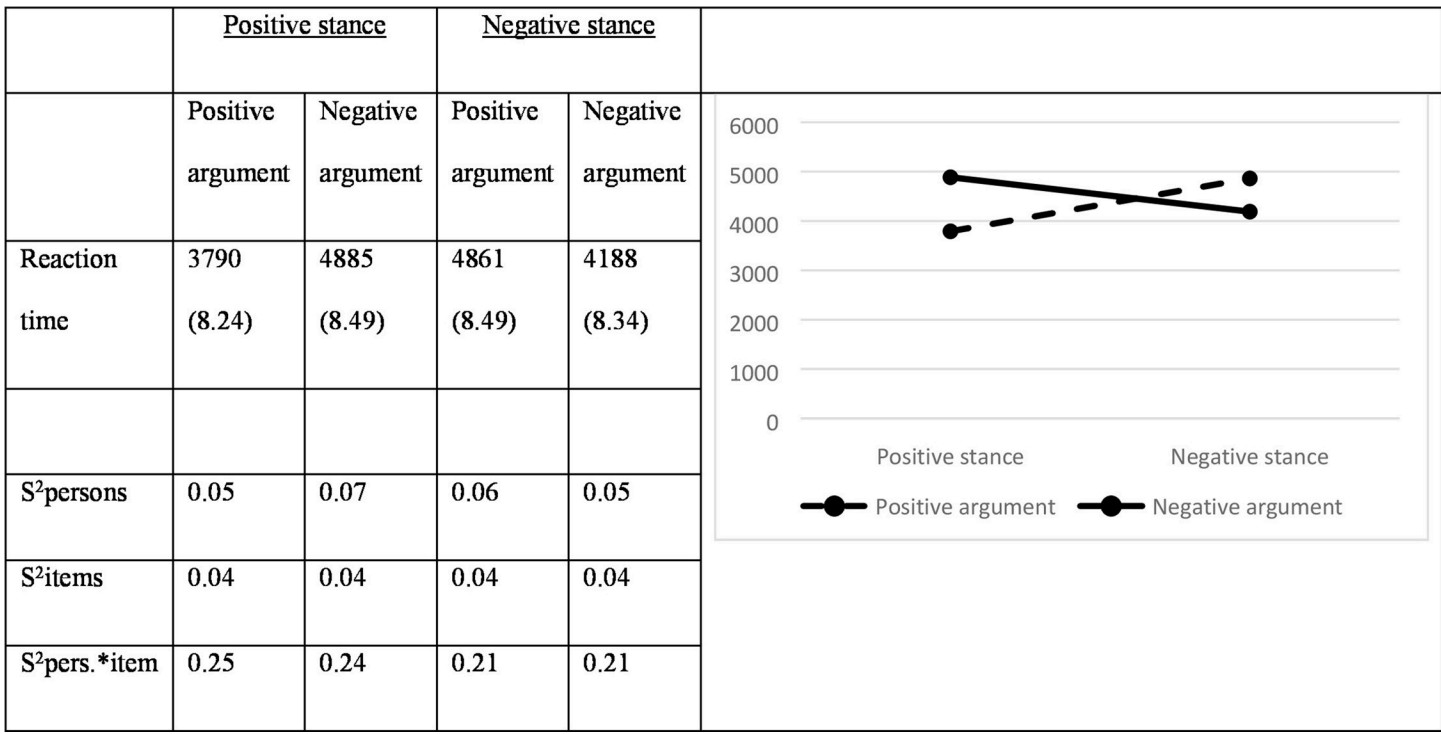

|  | Positive stance | | Negative stance | |
|---|---|---|---|---|
|  | Positive argument | Negative argument | Positive argument | Negative argument |
| Reaction time | 3790 (8.24) | 4885 (8.49) | 4861 (8.49) | 4188 (8.34) |
|  |  |  |  |  |
| $S^2$persons | 0.05 | 0.07 | 0.06 | 0.05 |
| $S^2$items | 0.04 | 0.04 | 0.04 | 0.04 |
| $S^2$pers.*item | 0.25 | 0.24 | 0.21 | 0.21 |

**Fig 4. Mean reaction times in milliseconds (Logs between brackets) and their variances (in Logs).**

## Conclusion and Discussion Study 2

The main goal of Study 2 was to see if markedness effects can be detected when stimuli have been controlled for word length and word frequency. Our results did not display an effect for the accuracy scores, but they did show an MP effect in the expected direction for the reaction times. This effect, however, was smaller in size than the MP effect found in Study 1. This suggests that the MP effect in Study 1 might well partially have been caused by word frequency, but irrespective of this, MP is an important factor in framing research.

In addition, Study 2 again shows results that are in line with predictions from AO. This principle predicts that speakers and writers choose arguments that–in terms of valence–are in line with the conclusion they aim to draw. When one is positive about a tennis player, their performance is described in terms of matches won rather than in terms of matches lost. AO predicts that readers and listeners pick up on these signals, which could explain why congruent pairs are judged faster and more accurately.

Studies 1 and 2 tested the prediction from AO that congruent stance-argument pairs are responded to faster and more accurately than incongruent ones. This outcome, however, is not unique for AO as the more general notions of lexical priming and spreading activation would make similar predictions. Lexical priming effects are explained as caused by spreading activation: upon reading or hearing words, one does not only activate the meaning of that specific word, but, due to spreading activation, also related terms (e.g., [25], [26], [27]). This explains why, for example, the word 'tennis' can be recognized more easily after having read the word 'deuce' rather than 'dice'. In a similar way, it could also be the case that the positive (or negative) word in the stance (*good*, *bad*) primes the positive (or negative) word in the argument (*win*, *lose*), see also [1]. The congruency effects in Study 1 and Study 2 could then be explained by this more general phenomenon.

In order to disentangle AO and lexical priming as explanations for the patterns in Study 1 and 2, a third experiment was conducted with an additional manipulation. We varied not only the polarity of the stance and the argument, but also the distance between the evaluative term in the stance and in the argument (see the methods section of Study 3 for an example). If the congruency effect is due to lexical priming, the distance between the two words is expected to influence the strength of the effect, with a longer distance resulting in attenuated effects. From an AO perspective, the distance should not influence the effect. Hence, priming predicts a three-way interaction between the distance between the evaluative terms, the valence of the stance and the valence of the argument, whereas AO only predicts a two-way interaction similar to the one found in Study 1 and 2.

## Method Study 3

### Design and materials

Study 3 expands on the design used in Study 1 and 2. Participants judged stance-argument pairs that differed not only in stance valence (positive or negative) and argument valence (positive or negative), but also in the distance between argument and stance (adjacent or distant). The manipulation of the distance was achieved by adding a third, descriptive sentence to each item that was either the first sentence as in (a) or the sentence in between stance and argument as in (b). Because of this additional manipulation, the valence terms either occurred in consecutive sentences (a) or relatively far away from each other (b).

(a)  This student has taken courses in Linguistics and Visual Communication.

He has passed 90% of his exams.

He is smart.

(b)  This student has passed 90% of his exams.

He has taken courses in Linguistics and Visual Communication.

He is smart.

Please note that the introduction of this additional manipulation lead to a change in the structure of the experimental materials, as we now presented the argument first, and the stance thereafter; Study 1 and 2 presented the stance before the argument similar to what has been done in some of the experiments reported in [2]. Using this structure for experiment 3, however, would lead participants to interpret the sentence added to create distant and adjacent conditions (in the example: "he has taken courses in Linguistics and Visual Communication") as an additional argument for the stance.

Participants judged 64 items in total, related to 8 scenarios that each occurred in 2 (stance valence: positive or negative) x 2 (argument valence: positive or negative) x 2 (distance: adjacent or distant) versions. Moreover, and just like in Study 1 and 2, participants judged an equal number of 'false' items to prevent automatic responding. The full list of experimental materials has been provided in S4 Appendix.

### Participants

Participants ($N = 60$) were students at Tilburg University. They received partial course credit for participation. Most of the participants were female ($N = 41$) and their mean age was 21.8 ($SD = 4.1$). Participants were only allowed to take part in Study 3 if they had not already participated in Study 1 or 2.

## Results Study 3

### Accuracy

The accuracy scores for Study 3 are displayed in Fig 5. Whereas there are no main effects for the argument valence ($\chi^2 = 1.87$; df = 1; $p = .17$) or the distance between stance and argument ($\chi^2 = 0.01$; df = 1; $p = .91$), there is a very small-sized main effect of stance valence: stimuli containing negative stances were verified correctly more often than positive ones ($\chi^2 = 8.27$; df = 1; $p < .01$; Cohen's $d = 0.04$). Once again, there is an interaction between stance and argument ($\chi^2 = 24.35$; df = 1; $p < .001$): positive stances are more often judged correctly when they are preceded by a positive argument ($\chi^2 = 9.27$; df = 1; $p = .002$; Cohen's $d = 0.30$), whereas negative stances are more often judged correctly when they are preceded by a negative argument ($\chi^2 = 26.01$; df = 1; $p < .001$; Cohen's $d = 0.34$), thus supporting the Argumentative Orientation Hypothesis. The other relevant two- and three-way interactions failed to reach significance (in all cases $\chi^2 < 0.58$; df = 1; $p > .44$). This means that the interaction between stance and argument valence occurs irrespective of whether stance and argument are presented in consecutive sentences or separated by a third sentence.

To test for MP effects, we compared the positive and negative congruent pairs. Interestingly, we found that verifying negative congruent pairs is done more accurately than verifying positive congruent pairs ($\chi^2 = 6.52$; df = 1; $p = .01$, Cohen's $d = 0.14$). Hence, there is an effect of MP, but it is in a different direction than expected.

### Reaction times

The RT are displayed in Fig 6. Whereas there are no main effects for the stance valence ($\chi^2 = 3.02$; df = 1; $p = .08$) or distance ($\chi^2 = 0.99$; df = 1; $p = 0.32$), there is a very small-sized main effect of argument valence: positive arguments are processed slower than negative arguments ($\chi^2 = 12.53$; df = 1; $p < .001$; Cohen's $d = 0.04$). This main effect, however, should be interpreted in light of the interaction between argument valence and stance valence ($\chi^2 = 58.16$; df = 1; $p < .001$): positive stances go well together with positive arguments ($\chi^2 = 17.60$; df = 1; $p < .001$; Cohen's $d = 0.29$) while negative stances are best defended using a negative argument ($\chi^2 = 48.13$; df = 1; $p < .001$; Cohen's $d = 0.28$).

In addition, we observed a two-way interaction between stance valence and distance ($\chi^2 = 4.19$; df = 1; $p = .04$). However, none of the pairwise comparisons reaches significance: stimuli with a positive stance and intervening sentence before the argument show a trend in longer RT ($\chi^2 = 3.22$; df = 1; $p = .07$), whereas this is not the case for negative stances ($\chi^2 = 0.05$; df = 1; $p = .83$). The two-way interaction between stance valence and distance failed to reach significance ($\chi^2 = 0.02$; df = 1; $p = .88$), as did the predicted three-way interaction between argument valence, stance valence and distance ($\chi^2 = 0.001$; df = 1; $p = .97$). This again implies that congruency effects occur irrespective of the distance between stance and argument.

To test for MP effects, we compared the positive and negative congruent pairs. Similar to what we have seen for the accuracy scores in Study 3, there is an MP effect but it is in an unexpected direction: fully negative pairs are verified more quickly than fully positive pairs ($\chi^2 = 14.96$; df = 1; $p < .001$, Cohen's $d = 0.21$).

## Conclusion and Discussion Study 3

Study 3 contained an expanded design compared to Study 1 and 2 in order to see whether lexical priming might have caused the effects in those studies. The RT and accuracy scores once again showed two-way interaction effects, as predicted by Argumentative Orientation. However, no three-way interaction with distance was observed. If lexical priming causes the

| | Positive argument | | Negative argument | | |
|---|---|---|---|---|---|
| | Positive stance | Negative stance | Positive stance | Negative stance | |
| Adjacent | | | | | 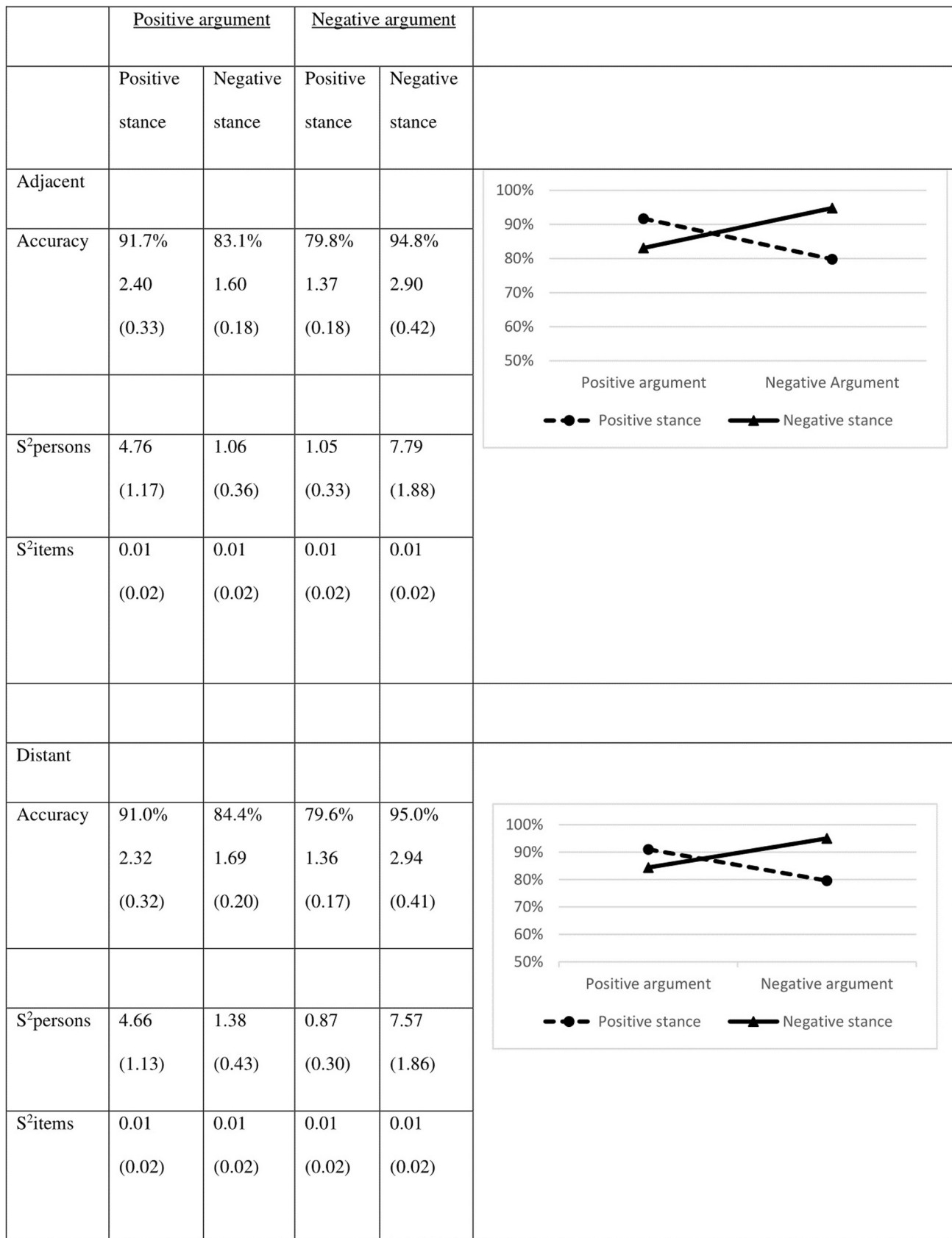 |
| Accuracy | 91.7% 2.40 (0.33) | 83.1% 1.60 (0.18) | 79.8% 1.37 (0.18) | 94.8% 2.90 (0.42) | |
| | | | | | |
| S²persons | 4.76 (1.17) | 1.06 (0.36) | 1.05 (0.33) | 7.79 (1.88) | |
| S²items | 0.01 (0.02) | 0.01 (0.02) | 0.01 (0.02) | 0.01 (0.02) | |
| | | | | | |
| Distant | | | | | |
| Accuracy | 91.0% 2.32 (0.32) | 84.4% 1.69 (0.20) | 79.6% 1.36 (0.17) | 95.0% 2.94 (0.41) | |
| S²persons | 4.66 (1.13) | 1.38 (0.43) | 0.87 (0.30) | 7.57 (1.86) | |
| S²items | 0.01 (0.02) | 0.01 (0.02) | 0.01 (0.02) | 0.01 (0.02) | |

**Fig 5. Mean accuracy scores in percentages (Logits between brackets) and their variances (in Logits).**

congruency effects, these effects should have been stronger for adjacent stance-argument pairs than for distant pairs. Hence, Study 3 suggests that the congruency effects are caused by AO rather than solely by linguistic priming.

Surprisingly, the pattern observed here for congruent positive vs negative pairs is contradictory to the general Markedness Principle: the pairs containing negative words were processed more accurately and more quickly. This raises the question what caused these unexpected effects to arise. We discuss some ideas and suggestions on how to test these in future research in the general discussion.

## Overall Conclusion and Discussion

The three studies show strong and consistent effects of valence congruency: processing a stance-argument pair goes faster if the stance and argument are either both positive (*good-win*) or negative (*bad-lose*), compared to incongruent pairs (*good-lose* or *bad-win*). There is not just a facilitating effect in terms of processing speed: deciding whether the argument is in line with the stance is also done more accurately for congruent pairs than for incongruent pairs. Although the congruency effects are always statistically meaningful, they are particularly strong, that is: statistically large, for accuracy scores. Moreover, these effects occur independent of word length and word frequency, and also irrespective of the distance between stance and argument. Altogether, these findings indicate that the congruency effects are indeed likely to be primarily caused by AO, rather than by other mechanisms such as word frequency and lexical priming.

A point of discussion is whether it is reasonable to exclude lexical priming as the (main) source of the congruency effects in these studies. Even in the Distant conditions in Study 3, participants saw the stance and the argument sentences on the same screen. In a future study, we therefore suggest that new attempts should be made to distinguish AO and lexical priming effects. One way to tease these explanations apart would be to design experimental materials that are not susceptible to linguistic priming effects. In these studies, the arguments could for example be presented visually rather than textually (e.g., a graph showing 9 out of 10 green balls vs. 1 out of 10 red balls representing matches won/lost respectively) and respondents might then be asked to verify a stance (e.g., *the tennis player is good*) taking into account the match results. If a positive stance is verified faster after seeing more green than red balls, this provides strong evidence for AO effects independent of lexical priming.

A second aim of our research was to explore effects of markedness. In our interpretation of the Markedness Principle, we predicted that negative terms would be processed more slowly and less accurately than positive terms, as the latter generally constitute the unmarked item in contrastive pairs (e.g. *win-lose*, *good-bad*). This prediction was born out for Study 1 and 2, although the effects were statistically small. The results of Study 2 suggest that word frequency, a potential confound for (un)markedness, is not the only driver of the markedness effect. Yet, Study 3 showed a reversed pattern, albeit with statistically very small effect sizes: here we saw that the negative congruent pairs were judged more accurately and quicker than positive congruent pairs. This raises the questions as to *why* the results in Study 3 are not in line with our predictions and the results in Study 1 and 2.

The experimental materials used in Study 3 are different in at least two ways from those in Study 1 and 2. First, in Study 3 our materials covered slightly longer discourse contexts as a third descriptive sentence was added to the stance-argument pairs. Second, in Study 3 the argument was presented first, followed by the stance, whereas in 1 and 2 this order was reversed. Both of these changes contribute to a potentially more context-specific effect of MP. We elaborate on this contextual notion of markedness first, before explaining why the

| | Positive argument | | Negative argument | | |
|---|---|---|---|---|---|
| | Positive stance | Negative stance | Positive stance | Negative stance | |
| Adjacent | | | | | 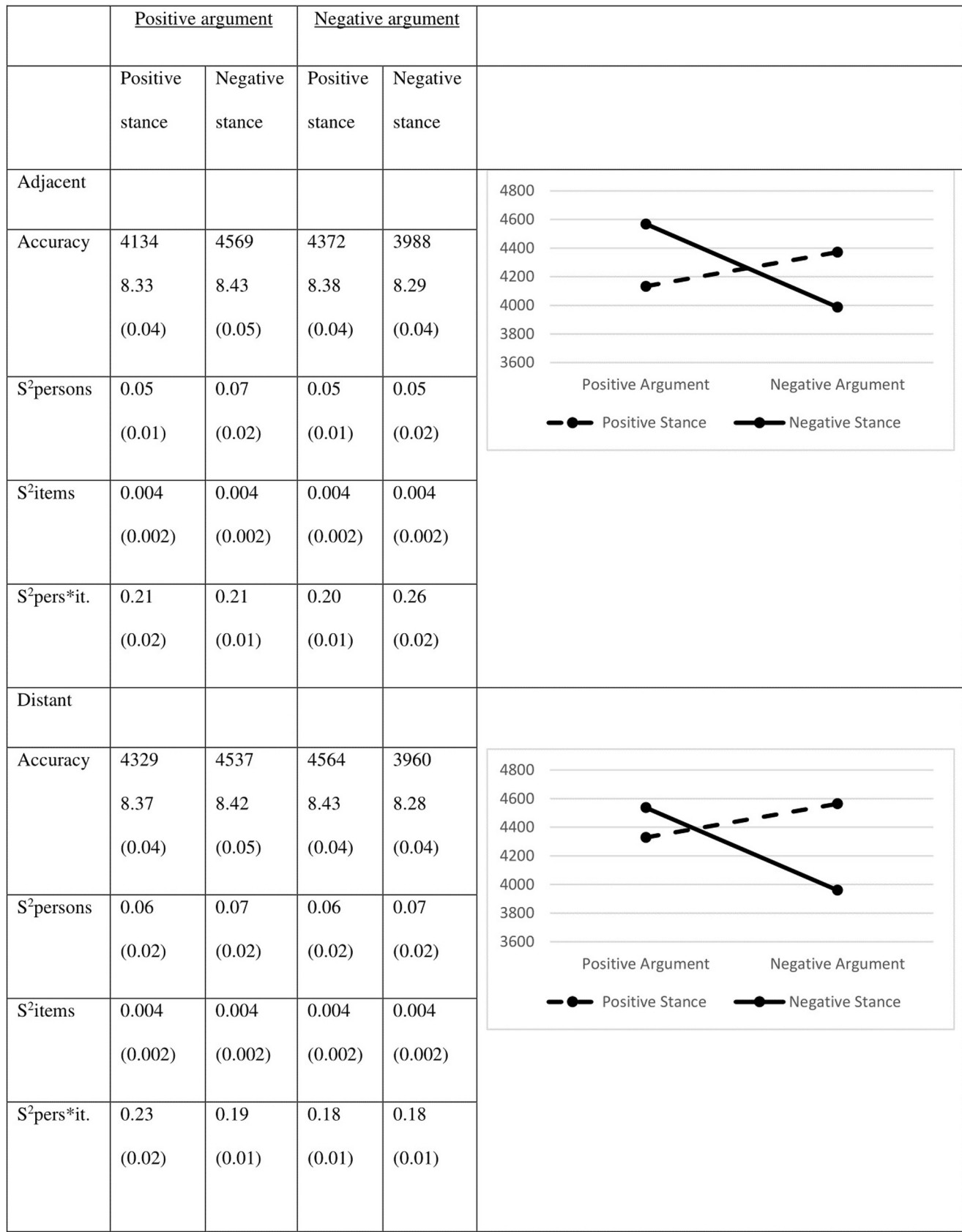 |
| Accuracy | 4134 | 4569 | 4372 | 3988 | |
| | 8.33 | 8.43 | 8.38 | 8.29 | |
| | (0.04) | (0.05) | (0.04) | (0.04) | |
| $S^2$persons | 0.05 | 0.07 | 0.05 | 0.05 | |
| | (0.01) | (0.02) | (0.01) | (0.02) | |
| $S^2$items | 0.004 | 0.004 | 0.004 | 0.004 | |
| | (0.002) | (0.002) | (0.002) | (0.002) | |
| $S^2$pers*it. | 0.21 | 0.21 | 0.20 | 0.26 | |
| | (0.02) | (0.01) | (0.01) | (0.02) | |
| Distant | | | | | |
| Accuracy | 4329 | 4537 | 4564 | 3960 | |
| | 8.37 | 8.42 | 8.43 | 8.28 | |
| | (0.04) | (0.05) | (0.04) | (0.04) | |
| $S^2$persons | 0.06 | 0.07 | 0.06 | 0.07 | |
| | (0.02) | (0.02) | (0.02) | (0.02) | |
| $S^2$items | 0.004 | 0.004 | 0.004 | 0.004 | |
| | (0.002) | (0.002) | (0.002) | (0.002) | |
| $S^2$pers*it. | 0.23 | 0.19 | 0.18 | 0.18 | |
| | (0.02) | (0.01) | (0.01) | (0.01) | |

**Fig 6. Reaction times in milliseconds (Logs between brackets) and their variances (in Logs).**

structure of the experimental materials in Study 3 has probably triggered this different version of MP to play a large role.

Several authors ([2], [12], [28]) have argued context can overrule the general notion of markedness that was discussed in the theoretical framework for this paper (positive terms are generally unmarked and therefore easier to process): although negative terms may be marked by default, negative terms can be unmarked in language use contexts. To illustrate this, image a situation in which Person A has just had a root canal treatment, and meets with his friend Person B. In such a situation, it will be less marked for B to ask "*How bad was your root canal treatment*?*"* rather than asking how *good* it was. The strong negative expectation about the "fun" of root canal treatments in this case causes the negative term to be the unmarked frame to use for asking a question.

A consequence of this contextual notion of markedness is that negative terms carry a more restricted pragmatic interpretation than their positive counterparts do. This is because positive terms are used both in contexts that do not specify an a priori frame and in situations that clearly ask for a positive frame, while negative terms are only used if there is a good reason to do so (e.g., a negative expectation). Following this line of reasoning, a contextual notion of markedness may predict congruent negative pairs to be processed faster and more accurately than congruent positive pairs, because in the incremental process of reading a sentence the negative first part triggers a negative continuation more strongly than a positive "default" part triggers a positive continuation (compare: [2]).

So, why would we see this contextual-version of the MP principle in Study 3 and not in Study 1 and 2? We think that this is because expectations about the valence of the second part of the stimuli played a more significant role in the argument-stance order in Study 3, than in the stance-argument order in Study 1 and 2; there must be a stance after an argument whereas stances are not necessarily backed up by arguments. Second, the contextual notion of MP is more likely to reveal itself in larger discourse contexts, and therefore, in Study 3. To disentangle the possible relation between the order of stance and argument, the length of the experimental stimuli and markedness effects we propose further studies in which (a) the same experimental materials are presented in both orders (stance-argument and argument-stance) and (b) the same experimental materials are used in smaller or longer discourse contexts. Such future studies are not only relevant for advancing our understanding of the results found in the current research, but it is also relevant in relation to a broader understanding of what markedness really is.

Another point of discussion concerns the external validity of our studies. In all three studies, the samples were relatively homogeneous: young, higher-educated participants, the majority of them women. In earlier work, [2] found the effect of valence framing to be consistent across student (Studies 1, 2, 3, and 5) and non-student (Studies 4 and 6) samples, as well as for male and female participants. Therefore, we have no *a priori* reason to assume that results observed in our studies would be different for participants with different educational backgrounds, or that gender plays a role in these effects. However, for another type of framing, risky choice framing, it has been found that some sample characteristics may play a role, such as gender [29]. We would welcome attempts to replicate our findings with other populations.

Overall, our results are in line with predictions from AO: readers find it easier to verify congruent stance-argument pairs than to verify incongruent pairs and they are also faster in their judgments. Hence, it may well be that the valence-consistent shift that is observed in an abundance of framing studies is caused by readers picking up on the (hidden) arguments in the words of the speaker or writer. Not only is this theoretically interesting, but it is also a relevant finding for the practice of writing argumentative texts; writers can best use arguments that share the valence of the standpoints they aim to defend, as they are easier and quicker to process.

## Supporting information

**S1 Appendix. Experimental materials Study 1.**
(DOCX)

**S2 Appendix. Formalization of the multi-level models used.**
(DOCX)

**S3 Appendix. Experimental materials Study 2.**
(DOCX)

**S4 Appendix. Experimental materials Study 3.**
(DOCX)

## Acknowledgments

We are grateful to Hanneke van der Kallen, Felix Broekhuizen, and Marie Barking who collected the data in Studies 1 (Fall 2015), 2 (Fall 2016), and 3 (Winter 2018) respectively, as part of the master's thesis and research masters' traineeship. Studies 1 and 2 were presented at the 2017 *Etmaal van de Communicatiewetenschap* and were reported on in Dutch in the *Tijdschrift voor Communicatiewetenschap* (TvC). We are grateful to the audience at the *Etmaal* and to the reviewers and editors at TvC for the comments and feedback.

## Author Contributions

**Conceptualization:** Naomi Kamoen, Maria Baukje Johanna Mos.

**Data curation:** Maria Baukje Johanna Mos.

**Formal analysis:** Maria Baukje Johanna Mos.

**Investigation:** Naomi Kamoen, Maria Baukje Johanna Mos.

**Methodology:** Naomi Kamoen, Maria Baukje Johanna Mos.

**Project administration:** Naomi Kamoen.

**Supervision:** Naomi Kamoen, Maria Baukje Johanna Mos.

**Validation:** Naomi Kamoen.

**Writing – original draft:** Naomi Kamoen, Maria Baukje Johanna Mos.

**Writing – review & editing:** Naomi Kamoen, Maria Baukje Johanna Mos.

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
