## [Decision Letter · Decision Letter 0]

26 Sep 2019

PONE-D-19-24361

A Good Tennis Player Does Not Lose Matches

The Effects of Valence Congruency in Processing Stance-Argument Pairs

PLOS ONE

Dear Dr. Kamoen,

Thank you for submitting your manuscript to PLOS ONE. After careful consideration, we feel that it has merit but does not fully meet PLOS ONE’s publication criteria as it currently stands. Therefore, we invite you to submit a revised version of the manuscript that addresses the points raised during the review process.

Please find below the reviewers' comments.

We would appreciate receiving your revised manuscript by Nov 10 2019 11:59PM. To enhance the reproducibility of your results, we recommend that if applicable you deposit your laboratory protocols in protocols.io, where a protocol can be assigned its own identifier (DOI) such that it can be cited independently in the future. For instructions see: http://journals.plos.org/plosone/s/submission-guidelines#loc-laboratory-protocols

We look forward to receiving your revised manuscript.

Kind regards,

Valerio Capraro

Academic Editor

PLOS ONE

Journal Requirements:

Additional Editor Comments (if provided):

I have now collected two reviews from two experts in the field. The reviewers are positive, but made several comments to improve the paper. Therefore, I would like to invite you to revise your paper following the reviewers' comments. I am looking forward for the revision.

Reviewers' comments:

Reviewer's Responses to Questions

**Comments to the Author**

1. Is the manuscript technically sound, and do the data support the conclusions?

Reviewer #1: Yes

Reviewer #2: Yes

2. Has the statistical analysis been performed appropriately and rigorously? 

Reviewer #1: Yes

Reviewer #2: Yes

3. Have the authors made all data underlying the findings in their manuscript fully available?

Reviewer #1: Yes

Reviewer #2: No

4. Is the manuscript presented in an intelligible fashion and written in standard English?

Reviewer #1: Yes

Reviewer #2: Yes

5. Review Comments to the Author

Reviewer #1: This manuscript presents results of three related studies testing for potential effects outlined by the Argumentative Orientation (AO) and Markedness Principle (MP). The description of the AO and MP are solid, even for a non-expert, and characterize why one might, for example, consistent with the AO, hypothesize that congruency (e.g., positive-positive or negative-negative rather than positive-negative or negative-positive) between stance-argument pairs may be differentially recalled (accuracy and reaction time measures). The findings overall show support for AO, but mixed support for MP.

The manuscript is well-written and structured and conveys well the key findings, plus places them in the context of leading theories. I would only raise several considerations about whether the manuscript should be published, given PLOS ONE guidelines. The samples are fairly small and narrow in age and background (sample sizes of 68, 97 and 60 young Dutch university adults in early 20s, the majority of them women). The sample characteristics should be more explicitly highlighted as a limitation to generalizability across variable human ages and more diverse international and socioeconomic samples. Are there ages (e.g., evidence that older adults tend to have more positive outlook than younger adults) or circumstances (e.g., people living in more food and water and housing insecurity possibly making more rapid and accurate links between negative valences compared to others living with fewer such challenges?) for which findings might be anticipated to differ, but that are not covered in this narrow demographic sample?

Please further discuss in another paragraph in the Discussion the effect sizes observed, and what those might mean about the importance of the effects tested and observed. With respect to ethical approval, I don't know what regulations might apply to the Netherlands at the time the studies were conducted; it's noted that individual informed consent was obtained but apparently that institutional ethical approval was not necessary. Please clarify if/whether not having institutional ethical approval is/was consistent with national guidelines and international best practices at that time.

Reviewer #2: Kamoen and Mos conducted three studies designed to explore the argumentative orientation (AO) and markedness principles. Briefly, the basic design of all three studies involved having participants respond to the logic of congruent and incongruent pairs of sentences (e.g., The tennis player is good. She won 20 of 25 matches. vs. The tennis player is good. She won 5 of 20 matches). The results support the AO principle, even under conditions designed to test alternative explanations such as lexical priming (study 3). The markedness principle was supported in studies 1 and 2, but not in study 3, where the effect was reversed. The authors suggest some potential explanations to account for this.

I should note that the research presented in this manuscript is outside of my area, so it is likely that I would have missed any mistakes or details related to the theoretical motivations for the study. Keeping that in mind, I found the experimentation and reporting of the studies to be excellent. The methodology appears to be solid, and the authors very rigorous in the application of the study. The write-up is very good and I really only have some minor comments/questions.

- I am a little concerned about the ethics statement. As far as I know, IRB approval is not contingent on whether or not participants are part of a vulnerable population. We still need ethical approval to conduct experimentation on human subjects regardless of what population they belong to.

- This likely falls beyond the scope of the work presented, but I was wondering about statements that appear negative, but are in a positive context. He is a good tennis player. He has only lost 5 of his last 25 matches. Does this make any difference?

- Why was the Dutch report anonymized? Also in the acknowledgments? Was this paper previously sent to a journal that has anonymous reviewing?

- Institute is used as the location for where subjects were recruited. This will need to be updated (see above point on this).

- I think the decimal points could be dropped in the tables (the actual figure only, on the y-axis)

- Is there a possible ceiling effect?

- In study 2, why is there such a drop in performance when a positive stance is followed by a negative argument (61.5%) when compared with a negative stance followed by a positive argument (76.7%). The RTs do not follow this. Why not?

- Reference 32 in the reference section is not complete.

6. PLOS authors have the option to publish the peer review history of their article (what does this mean?). If published, this will include your full peer review and any attached files.

Reviewer #1: No

Reviewer #2: No

---

## [Author Response · Author response to Decision Letter 0]

14 Oct 2019

Reviewer 1

1. Reviewer 1 asks us to reflect on the relatively homogeneous samples in the three studies and to highlight this limitation of the external validity by discussing which findings may or may not be expected to differ in other samples of respondents. 

We think that the reviewer is right that our three studies include very homogeneous samples of young, higher-educated participants, the majority of them women. We have now explicitly mentioned these characteristics of the sample in the participant sections of the three studies and we have indicated that these kind of homogeneous samples occur frequently in framing studies (compare for example the various studies in Holleman & Pander Maat, 2009). Moreover, we have included a new paragraph in the discussion section in which we elaborate on the extent to which the bias in the sample might hamper the generalizability of the results.

2. The reviewer also asks to discuss the effect sizes of the reported effects in the discussion section. We have added mentions to the statistical effect sizes for the congruence and markedness effects in the discussion section. There are different ways to calculate effect sizes; we have opted for the more conservative option here, to classify the size of the effect relative to the combined between person and between item standard deviation (Cohen’s d). Using this calculation, results across the 3 studies generally show statistically large (i.e., Cohen’s d > .8) effects of congruency for the accuracy scores and results generally show small effects for markedness. We have now highlighted this pattern in our discussion paragraph. 

3. Reviewer 1 asks to clarify if/whether not having institutional ethical approval is/was consistent with national guidelines and international best practices at that time.

On May 25 2018, the General Data Protection Regulation (GDPR) entered into force in the Netherlands. From that point in time, our department at Tilburg University obliged researchers to obtain approval by an ethical committee before fielding any study with human participants. Our studies, however, have been conducted before the GDPR came into effect as Study 1 was run in the Fall of 2015, Study 2 in the Fall of 2016 and Study 3 in the Winter of 2018. At that time, review by an ethical committee was not considered a necessary step before fielding a scientific study with human participants, as long as the population of the study did not include participants under the age of 16 or participants that are part of a vulnerable population. Although we did not apply for ethical approval by an ethics committee, we would like to highlight that our participants did sign a consent form in which they consented for recording reaction times and responses and for the usage of these data in anonymized form for scientific research. Furthermore, the coordinator of the student participant pool (Jacqueline Dake) approved the request form and consent forms for all three studies. We have uploaded the consent forms used in the studies to Dataverse, where all data files can also be found. The handle for this Dataverse environment is https://hdl.handle.net/10411/SFFENZ

Reviewer 2:

1. Reviewer 2 also made comments about our ethics statement. We would like to refer to our explanation for the third point raised by reviewer 1.

2. Reviewer 2 makes an interesting observation about statements that appear negative, but are in a positive context, e.g., “He is a good tennis player. He has only lost 5 of his last 25 matches”. We agree that the addition of ‘only’ is likely to make a difference. In our experimental stimuli, we strived for minimal differences between conditions, and therefore did not include any adverbial markers such as ‘only’. It may well be, that words like ‘only’ serve to explicitly mark the incongruency in the sentence pair, in a way: the speaker highlights why in a positive evaluation (‘he is a good tennis player’) mention is made of a negative event (games that were lost): there were only few of them, making a remark about this negative event compatible with the positive evaluation. While this does, as the reviewer already notes, fall outside the scope of this particular manuscript, we think it could be empirically tested whether such incongruency markers are effective in a reaction time study (where the hypothesis would be that these markers facilitate processing of incongruent sentence pairs). A corpus study could also shed light on the extent to which these incongruent pairs, especially those with a positive stance and a negatively formulated argument which are expected to occur relatively rarely, tend to be marked with these types of adverbial phrases (‘only’, ‘no more than’, ‘just’, etc.). 

3. Reviewer 2 wonders why some information in our manuscript was anonymized (e.g., the acknowledgments, the name of the institute were the studies were run etc.; combined point 3 and 4 by reviewer 2).This was a mistake. We have de-anonymized all information in the text and in the acknowledgment. 

4. Reviewer 2 notes that the decimal points on the y-axis in the visualization of the interactions in the tables could be dropped. We agree, and have removed them.

5. Reviewer 2 wonders if there is a possible ceiling effect. We take this comment to apply to the accuracy scores. These range between 61.5 and 98.8% correct (on a binary choice task). However, we do observe clear effects of the manipulated variables on the accuracy scores in all three studies. If there is in fact a ceiling effect, this seems to be present only for one (or perhaps two) of the conditions, thus not obscuring the effects of our independent variables. The high scores overall are also an indication that the task instructions were clear, and the task was not difficult, which is what we intended.

6. In study 2, why is there such a drop in performance when a positive stance is followed by a negative argument (61.5%) when compared with a negative stance followed by a positive argument (76.7%). The RTs do not follow this. Why not?

We agree with Reviewer 2 that this is surprising. The drop in performance for stimuli with a positive stance followed by a negative argument is similar to the pattern in accuracy score for Studies 1 and 3, if more pronounced. We cannot account for the fact that the reaction times do not show the same pattern here – numerically, the condition with positive stance + negative argument is the slowest, but the difference with the other incongruent condition (negative stance + positive argument) is small. As we do not want to engage in speculations, it concerns the absence of an effect, and the effect size for effects in terms of accuracy scores is much larger in each case than that for reaction times, we feel that this is not a contradictory result, compared to the other studies. We have therefore not discussed it in the discussion section. 

7. Reviewer 2 has made us aware that reference 32 was not complete; we have updated this reference. 

Data transparency

All datafiles, consent forms, and additional information on the word frequencies of the stimulus materials in all three studies have been uploaded to Dataverse where they can be accessed. The handle for accessing these files is https://hdl.handle.net/10411/SFFENZ. The Dataverse environment has been scanned by Dataverse and has been approved for publication.

---

## [Editor Report · Decision Letter 1]

16 Oct 2019

A Good Tennis Player Does Not Lose Matches

The Effects of Valence Congruency in Processing Stance-Argument Pairs

PONE-D-19-24361R1

Dear Dr. Kamoen,

We are pleased to inform you that your manuscript has been judged scientifically suitable for publication and will be formally accepted for publication once it complies with all outstanding technical requirements.

With kind regards,

Valerio Capraro

Academic Editor

PLOS ONE
---

## [Editor Report · Acceptance letter]

25 Oct 2019

PONE-D-19-24361R1 

A Good Tennis Player Does Not Lose Matches. The Effects of Valence Congruency in Processing Stance-Argument Pairs 

Dear Dr. Kamoen:

I am pleased to inform you that your manuscript has been deemed suitable for publication in PLOS ONE. Congratulations! Your manuscript is now with our production department. 

With kind regards,

on behalf of

Dr. Valerio Capraro 

Academic Editor

PLOS ONE